# Straw Incorporation with Exogenous Degrading Bacteria (ZJW-6): An Integrated Greener Approach to Enhance Straw Degradation and Improve Rice Growth

**DOI:** 10.3390/ijms25147835

**Published:** 2024-07-17

**Authors:** Xiaoshuang Wei, Wanchun Li, Ze Song, Shiwen Wang, Shujuan Geng, Hao Jiang, Zhenhui Wang, Ping Tian, Zhihai Wu, Meiying Yang

**Affiliations:** 1Faculty of Agronomy, Jilin Agricultural University, Changchun 130118, China; weixiaoshuang@jlau.edu.cn (X.W.); liwanchun2001@163.com (W.L.); 15886182088@163.com (Z.S.); wangshiwen991001@163.com (S.W.); gengshujuan2019@163.com (S.G.); jianghao@mails.jlau.edu.cn (H.J.); wzhjlau@163.com (Z.W.); tianping@jlau.edu.cn (P.T.); 2National Crop Variety Approval and Characterization Station, Jilin Agricultural University, Changchun 130118, China; 3College of Life Sciences, Jilin Agricultural University, Changchun 130118, China

**Keywords:** rice straw, physiological measurements, degradation efficiency, bacterial community, fungal community, yield

## Abstract

Rice straw is an agricultural waste, the disposal of which through open burning is an emerging challenge for ecology. Green manufacturing using straw returning provides a more avant-garde technique that is not only an effective management measure to improve soil fertility in agricultural ecosystems but also nurtures environmental stewardship by reducing waste and the carbon footprint. However, fresh straw that is returned to the field cannot be quickly decomposed, and screening microorganisms with the capacity to degrade straw and understanding their mechanism of action is an efficient approach to solve such problems. This study aimed to reveal the potential mechanism of influence exerted by exogenous degradative bacteria (ZJW-6) on the degradation of straw, growth of plants, and soil bacterial community during the process of returning rice straw to the soil. The inoculation with ZJW-6 enhanced the driving force of cellulose degradation. The acceleration of the rate of decomposition of straw releases nutrients that are easily absorbed by rice (*Oryza sativa* L.), providing favorable conditions for its growth and promoting its growth and development; prolongs the photosynthetic functioning period of leaves; and lays the material foundation for high yields of rice. ZJW-6 not only directly participates in cellulose degradation as degrading bacteria but also induces positive interactions between bacteria and fungi and enriches the microbial taxa that were related to straw degradation, enhancing the rate of rice straw degradation. Taken together, ZJW-6 has important biological potential and should be further studied, which will provide new insights and strategies for the appropriate treatment of rice straw. In the future, this degrading bacteria may provide a better opportunity to manage straw in an ecofriendly manner.

## 1. Introduction

China, as a traditional agricultural country in the world, produces more than 800 million tons of straw each year [1]. Such an enormous resource is often incinerated by traditional methods, which results in a large amount of smoke that causes serious problems with haze, while carbon dioxide (CO_2_) is emitted directly into the atmosphere, thus leading to global warming [2]. Straw return is an effective way to recycle straw resources, and it has the advantages of increasing soil fertility and regulating the soil microbial environment [3]. As an important renewable resource, crop straw is not only rich in carbon (C), nitrogen (N), phosphorus (P), potassium (K), and trace elements but also contains a large amount of organic components, such as lignin and cellulose [4,5,6]. Straw returning to the soil significantly improves soil quality indicators, such as the contents of major nutrients, soil structure, and biological functions [7,8,9]. However, straw is not easily degraded under natural conditions, which suggests that the application of straw returning processing technology can be severely limited [10,11,12,13]. Consequently, exogenous highly efficient bacterial agents that can efficiently degrade straw have been the subject of significant amounts of research.

Previous studies have shown that the addition of exogenous bacterial systems during straw incorporation would be more effective at degrading straw and causing significant differences in microbial communities [14]. The addition of exogenous composite bacterial systems can change the indigenous microbial community structure to some extent, which indicates that this process significantly increases the abundance and diversity of predominant bacteria and maintains long-term stability [15,16]. Recent studies have shown that exogenous composite microbial systems can be directly applied to incorporated straw. It has been reported that the incorporation of straw with the addition of microorganisms that can degrade lignocellulose accelerated the decomposition of straw in the soil and significantly increased the activity of indigenous soil microbes [17]. Chu et al. (2020) constructed combined bacteria and reported rates of degradation of corn (*Zea mays*) straw lignin, cellulose, and hemicellulose of 43.36%, 31.29%, and 48.36%, respectively, after 20 days under optimal conditions (32 °C) [18]. The same authors revealed that coordination among the microbial communities was key during the degradation of corn straw. PuentesTéllez and Falcao Salles (2018) developed minimal active microbial consortia (MAMC) that degraded corn stover up to 96.5% at 28 °C, aided by the functional diversity and metabolic complementarity of MAMC [19]. Compared with single strains, microbial consortia are more adaptable, have better interactions, and achieve more effective degradation. Qin et al. (2015) reported that the application of cellulose-decomposing bacteria (hereby defined as straw decomposers) could effectively hasten the process of straw decomposition and thereby improve the availability of soil nutrients [20].

Overall, the root causes of the differences in straw degradation are owing to effects on the soil microbial community [21]. Therefore, enhancing the efficiency of degrading straw by improving the structure of indigenous soil microbial communities are promising approaches. Soil bacteria and fungi are involved in a variety of critical ecological processes, and their diversity and community structure influence key soil characteristics, such as nutrient content, function, and stability [22]. Furthermore, key communities are reportedly strongly associated with soil function [23]. Nevertheless, the lack of suitable bacterial agents suggests that relatively few studies have been dedicated to their effects on indigenous soil microorganisms under straw return.

To accelerate the degradation of rice straw, we screened one strain in the early stage of the experiment that utilized ZJW-6, which has the ability to degrade stover. Accordingly, we investigated the effects of rice (*Oryza sativa*) straw returned to the field with the addition of ZJW-6 on the degradation of straw, content of soil nutrients, and rice yield. We also determined the effects of soil microorganism communities by utilizing Illumina-based 16S rRNA amplicon and ITS sequencing. The results of this research provide useful data to solve the problem of straw degradation in large-scale production for farmers, which is highly significant for accelerating the promotion of crop straw return to the field and achieving the sustainable use of agricultural resources. 

## 2. Results

### 2.1. The Impacts of Exogenous-Degrading Bacteria Treatments on the Morphological and Physiological Characteristics of Rice

Straw-degrading bacterial agents had important effects on the phenotypic growth of rice plants (Figure 1a). Compared with the control (S0), the S1 treatment significantly inhibited the growth of aboveground and belowground parts of the plant, while the S2 treatment caused different degrees of promotion and more vigorous root growth (Figure 1a). The effect of adding straw-degrading bacteria on the height of rice plants showed a tendency of increasing and then decreasing with the reproductive period and reached its highest value during the filling period. The S1 treatment significantly reduced the height compared with the other treatments, and the S0 treatment produced the highest plants (Figure 1b). The addition of straw-degrading bacteria could alleviate the negative effect of returning straw to the field. The effects of straw-degrading bacteria treatments on the leaf area of rice plants were consistent with the trend of plant height (Figure 1c). The results of the effect of different straw-degrading bacteria treatments on the tillering dynamics of rice plants are shown in Figure 1d. The number of tillers showed a trend of increasing and then decreasing with the period of fertility and reached their maximum 50 days after transplanting. There were significantly more tillers in the S2 treatment than in the other treatments during the late stage of tillering. Straw-degrading bacteria had significant effects on the accumulation of aboveground material in the rice in all periods (Figure 1e). The accumulation of aboveground dry matter showed a trend of increasing and then decreasing with the process of the period of fertility, and its peak appeared during the grouting period. The accumulation of dry matter in the S2 treatment showed an increasing trend throughout the whole period of fertility, and the S0 treatment maintained a higher accumulation of dry matter than S1. The difference was significant.

The addition of straw-degrading bacteria had a significant effect on the growth of rice roots, which showed a trend of increasing and then decreasing with the course of fertility (Figure 1f), with the peak occurring at the full heading stage. The application of straw-degrading bacteria increased the root activity, which demonstrated S2 > S0 > S1. The number of roots showed a trend of increasing and then decreasing (Figure 1g). The number of roots under the S2 treatment at maturity was significantly higher than that of the S1 treatment, and there was no significant difference between the S0 and S1 treatments, while S2 increased the number of roots by 7.75% compared with the S0 treatment. The S0 treatment resulted in significantly longer roots than the other treatments before the jointing stage. The root lengths showed a tendency to grow more quickly and then decrease throughout the reproductive period, with their maximum values occurring at the heading stage (Figure 1h). At the maturity stage, the plants in the S2 treatment were significantly higher than those of the S0 and S1 treatments, and there was no significant difference between the S1 and S0 treatments. The S2 treatment increased by 8.74% compared with the S0 treatment.

### 2.2. Effects of the Addition of Straw-Degrading Bacteria on the Photosynthetic Capacity of Rice

As shown in Figure 2a, the different treatments showed a trend of increasing and then decreasing the net photosynthetic rate and reached the maximum net photosynthetic rate at the full heading stage. The S2 treatment maintained a higher net photosynthetic rate throughout the whole period, which was significantly higher than that of the S0 treatment. The S0 treatment had higher stomatal conductance at the tillering stage (Figure 2b), but it was in a decreasing trend throughout the whole period. The S2 treatment showed a trend of increasing stomatal conductance, which then decreased, and reached its maximum value at the full heading stage. The concentration of intercellular CO_2_ of all the treatments was the highest at the tillering stage (Figure 2c), but there was no significant difference. The S2 treatment was higher than the other treatments in the full heading stage. The transpiration rates of the added straw-degrading bacterial agents all showed a trend of increasing and then decreasing and reached their highest values at the full heading stage (Figure 2d). In addition, the transpiration rate of the S2 treatment at the full heading stage increased by 78.7% compared with that at the tillering stage. The apparent mesophyll conductance of the S0 treatment was lower than that of the other treatments during the whole period (Figure 2e). The S2 treatment had higher levels of apparent mesophyll conductance at the tillering, jointing, and filling heading stages, which were significantly higher than those of the S0 and S1 treatments. The addition of straw-degrading bacteria resulted in a tendency of the SPAD value to increase and then decrease with the course of the reproductive period (Figure 2f), with its peak appearing at the full heading stage. There was no significant difference between the three treatments at the tillering, jointing, and booting stages, and S0 was significantly higher than S1 at the full heading stage. The S2 treatment was significantly higher than that of the other treatments at the filling stage. 

The increased photosynthetic capacity of the leaves will positively affect the yield. One year of straw return reduced the number of grains per spike in rice, which resulted in lower yields, and the addition of straw-degrading bacteria under straw return increased the rice yield by increasing the effective number of spikes and the 1000-grain weight (Appendix A).

### 2.3. Impacts on Rice Straw Degradation Rate 

The rate of straw decomposition continually increased after the transplantation of the rice seedlings. The straw degradation rate of the S2 treatment at the tillering stage was greater than that of S1, with an increase of 28.34%, and the straw degradation rate of the S2 treatment at the full heading stage was significantly higher than that of the S1 treatment by 11%. The straw degradation rate of the S2 treatment at the maturity stage was significantly higher than that of the S1 treatment by 15.42% (Figure 3a). Lignocellulose is primarily composed of hemicellulose, cellulose, and lignin. The degree of degradation of these three components characterizes the effect of straw degradation [24]. The addition of straw-degrading bacteria had a significant effect on the rate of degradation of the rice straw in different fertility periods. The rate of lignin degradation in the S2 treatment was 41.23%, 20.12%, and 17.42% higher than that of the S1 treatment at the tillering, full heading, and maturity stages, respectively (Figure 3b). Cellulose, as the main organic ingredient of straw, is inert, which renders it difficult for microorganisms to degrade. As shown in Figure 3c, the rate of degradation of cellulose of the S2 treatment at the tillering, full heading, and maturity stages was significantly higher than that of the S1 treatment by 31.21%, 14.23%, and 22.36%, respectively. This indicated that the addition of straw-degrading bacterial agents could effectively improve the degradation of cellulose and accelerate the mineralization of dry matter. The rate of degradation of hemicellulose in the S2 treatment was 32%, 14.38%, and 22.45% higher than that of the S1 treatment at the tillering, full heading, and maturity stages, respectively (Figure 3d).

### 2.4. SEM (Scanning Electron Microscopic) Observation on the Degradation of Straw

As shown by observation with SEM on the outer surface (Figure 4), the S2 treatment resulted in a thinner waxy silicified layer during the process of degradation, and the lower epidermis began to appear faintly until almost all the waxy silicified layer had disappeared. The cell structure was obviously damaged, and the complete cells could barely be seen in the field of view. In addition, the wax layer and cell structure of the straw epidermis were obviously collapsed, notched, and fractured. However, the straw structure was uniform in the S1 treatment, and the cells of the straw epidermis were evenly arranged with clear boundaries. The results of SEM observation of the straw further confirmed that the degrading bacterial agents were highly effective at causing degradation. This excellent characteristic of this bacterial agent is extremely beneficial to the efficient degradation of straw.

### 2.5. Soil Physicochemical Characteristics

The soil TN is a general term for various forms of N, which can be used to measure the degree of soil fertility. As shown in Figure 5a, the content of TN of the soil at maturity under each treatment showed a general trend of no change compared with that before returning the soil to the field. The content of AK in the soil showed a decreasing trend compared with that of the pre-return period, with the S0, S1, and S2 treatments decreasing by 26.38%, 25.66%, and 32.11%, respectively (Figure 5b). The S2 treatment showed the most obvious decrease in the content of AK. The content of AP in the soil showed an overall increasing trend compared with the pre-return period. The S0, S1, and S2 treatments increased by 29.04%, 38.97%, and 30.53%, respectively (Figure 5c). The straw return treatment increased the content of soil organic carbon (SOC), and there was no significant change in the S0 treatment (Figure 5d). The concentrations of SOC and AP clearly demonstrated that the treatments of straw incorporation and the addition of degradative bacterial agents provided greater soil fertility in comparison with the S0 treatment. 

### 2.6. Enzymatic Activity 

The data showed that the straw returning improved the activity of soil CAT (catalase). However, although the activities of ACP (acid phosphatase) and URE (urease) decreased, that of the SUC (sucrase) did not change significantly (Figure 6). The addition of degradative bacterial agents decreased the activity of soil CAT, while the activities of ACP, URE, and SUC increased.

### 2.7. Effects of Straw Return on the α-Diversity of the Soil Bacterial and Fungal Communities

This study utilized 16S rRNA and ITS rRNA to explore changes in the microbial communities. A total of 1,226,463 high-quality 16S rRNA gene sequences were obtained from the soil samples in this study. After equalizing the sampling effort, 16,960,597 to 20,154,106 sequences were retained and clustered into 8972 OTUs at 97% sequence similarity (Appendix A).

The degree of alpha-diversity was used to analyze the microbial community diversity of a sample. A larger Chao1 index indicated that there were more low-abundance species in the community, and a larger observed_OTUs index indicated that more species were observed. Larger pielou_e and Shannon indices indicated that the species were more uniformly distributed. The Observed_OTUs, Shannon and Chao1 indices and pielou_e increased significantly after the straw was returned to the field, and after the application of straw-degrading bacteria, they were significantly higher than those of the S0 and S1 treatments (Figure 7a). This indicated that the straw return significantly increased the diversity of bacteria in the soil, particularly when a straw-degrading bacterium was added. We also analyzed the diversity of fungal microorganisms in the soil (Figure 7b). The Observed_OTUs, Shannon, Chao1 indices, and pielou_e of the S2 treatment were lower than those of the S1 treatment. This indicated that the addition of straw-degrading bacterial agents reduced the degree of soil fungal microbial diversity.

### 2.8. Effect of Straw-Degrading Bacteria on the Microbial β-Diversity

An unconstrained principal coordinates analysis (PCoA) of the weighted UniFrac distance revealed that the bacterial and fungal communities varied among the treatments (PERMANOVA: *p* < 0.01; Figure 8). For the bacterial community structure, the PCoA analysis showed that PC1 explained 30.56% of the variation, and PC2 explained 17.64% of the variation. This result showed that there was difference in the composition of soil bacteria between the different treatment groups, such as S1 and S2, as compared with the control group S0, and the community composition of the S2 treatment was more extensive (Figure 8a). The cumulative percentage of the variance in the fungal communities in the samples explained by the PC1 was 36.47%, and the PC2 axes explained 31.28% of the variance in the fungal communities in the samples. The soil fungal communities formed three distinct clusters according to different treatments, compared with the control group S0 and S1, and the soil fungal composition of the S2 treatment groups showed significant differences (Figure 8b). These led to a significant separation of the microbial composition of fungi in the soil.

### 2.9. Microbial Community Structure and Composition

The bacterial community dynamics at the phylum level are shown in Figure 9. Proteobacteria, Acidobacterota, Firmicutes, Chloroflexi, Bacteroidota, Actinobacteriota, Gemmatimonadota, Myxococcota, Nitrospirota, and Verrucomicrobiota were the 10 most abundant phyla across all the samples (Appendix A). The main bacterial dominant taxa did not change after the straw return treatment; however, there was a significant increase in the abundance of Firmicutes, Acidobacterota, Gemmatimonadota, and Myxococcota. Some bacterial communities had significant decreases in the abundance of Nitrospirota, Chloroflexi, Verrucomicrobiota, and Bacteroidota. Actinomycetes and Bacteroidetes are considered crucial phyla to degrade lignocellulose [25]. An increase in their relative abundances can promote the degradation of lignocellulose. The S2 treatment was 23% more abundant in Bacteroidota than S1. Therefore, we hypothesized that straw degradation could be promoted by the application of degradative bacteria. The abundances of Bacteroidota and Nitrospirota increased significantly, while the abundance of Acidobacterota and Firmicutes decreased significantly after the application of straw-degrading bacteria. The abundances of Firmicutes and Gemmatimonadota increased significantly after the application of straw-degrading bacteria (S2) compared with the treatment without straw returning (S0). 

The variation in the bacterial communities was significant at the genus level. The top 10 bacterial dominant genera in terms of relative abundance in the different treatments were *Lactobacillus*, *Ellin6067*, *subgroup_7*, *KD4-96*, *Gemmatimona*, *Bacteroidetes_vadinHA17*, *sC--84*, *GOUTA6*, *Candidatus_Solibacter*, and *Muribaculaceae*; the main dominant bacterial taxa changed after the straw return treatment (Figure 9b, Appendix A), with a significant increase in the genus Muribaculaceae and a significant increase in the abundances of *sC--84* and *Lactobacillus*. The abundances of *Bacteroidetes_vadinHA17*, *GOUTA6*, and *KD4-96* decreased significantly. The abundances of *Lactobacillus*, *Gemmatimonas*, *SC-I-84*, and *Candidatus_Solibacter* increased significantly, and those of *Ellin6067*, *KD4-96*, *Bacteroidetes_vadinHA17*, and *GOUTA6* decreased significantly following the application of straw-degrading bacteria compared with the straw non-return treatment. Overall, significant (*p* < 0.05) differences were observed for the bacterial abundance among the treatments in different treatments at the genus level. These results reinforced that ZJW-6 indirectly promoted the degradation of straw by altering the bacterial community.

The fungal community composition was also altered by treatments with the straw-degrading bacteria (Figure 9c). Among all the samples, the seven most abundant phyla of fungi were Basidiomycota, Mortierellomycota, Ascomycota, Chytridiomycota, Rozellomycota, Olpidiomycota, and Glomeromycota (Appendix A). The addition of straw-degrading bacteria significantly decreased the relative abundances of Ascomycota, Olpidiomycota, and Glomeromycota but significantly increased those of Basidiomycota, Mortierellomycota, and Rozellomycota compared with the S1 treatment. The straw-degrading bacteria affected the composition of the fungal community at the genus level to varying degrees (Figure 9d, Appendix A). Under the experimental conditions of this study, the increase in the relative abundances of *Mortierella*, *Tausonia*, and *Mrakia* after the application of straw-degrading bacteria compared with that of S1, but the decrease in the relative abundances of *Fusarium*, *Aspergillus*, *Gibberella*, *Schizothecium*, *Massarina*, *Trichocladium*, and *Cladosporium*, and *S2* showed significant differences compared with those in S1 (*p* < 0.05). The complex microbial interactions caused by the addition of straw-degrading bacteria might promote the different bacterial and fungal communities to play different roles, which could improve the functional efficiency of microbes and promote their rates of degradation.

### 2.10. Regulatory Effects of Straw Incorporation and Degradative Bacteria on the Properties of Soil

SEM observations can offer an intuitive graphical representation to visualize the dynamic interactions between variables through fitting data to the models, which represents an arbitrary assumption [26]. It was utilized to investigate how the addition of exogenous degrading bacteria would affect the properties of straw-degrading and soil (Figure 10). Logically, the addition of ZJW-6 affected the growth of roots by indirectly regulating the soil microbial characteristics through their effects on straw decomposition and the soil physical and chemical properties. The results of a PLS-PM analysis (Figure 10) demonstrated that ZJW-6 improved the activity of enzymes (0.899, path coefficient, same below) and the rate of degradation (0.994, path coefficient), and the straw degradation regulates the soil microbial characteristics. Enzyme activity affected soil properties and increased the root number directly. The decomposition of straw affected the soil microbial α-diversity (Chao1, observed_OTUs, and pielou_e of the bacterial and fungal communities (0.507, -0.750, respectively). Both path coefficients were not significant. 

The above results showed that ZJW-6 not only directly participated in straw degradation but also stimulated the competitions of bacterial communities, which indirectly promoted straw degradation. The ZJW-6 also increased the root biomass by increasing the rate of degradation and activity of the soil enzymes, which negatively affected the effect of straw returning on plant growth, providing favorable conditions for rice growth, promoting its growth and development.

## 3. Discussion

Straw incorporation is a common practice to improve the soil fertility of rice paddies and the yield of rice [27,28,29]. A global meta-analysis showed that straw incorporation increased the soil organic matter and nutrients [29]. Indeed, straw incorporation also has some negative effects on the growth of rice plants. The incorporation of crop straw under anaerobic soil conditions can increase phytotoxic substances [30], thus inhibiting the growth of rice plants. In this study, the addition of straw-degrading bacterial agent ZJW-6 significantly alleviated the inhibition of growth and the development of the rice by returning straw to the field. The root vitality, root number, and root length were all significantly higher under S2 than under S1. There are studies reporting that straw mulching delayed the flag leaf senescence, which may support higher photosynthesis capacity during the grain filling period and have positive effects on yield [31,32]. This experiment showed that S2 treatment manifested itself in photosynthesis by promoting the opening of stomata in plant leaves, enhancing transpiration, improving the CO_2_ supply capacity of the leaf pulp cells and increasing the photosynthetic rate of leaves. The straw-degrading bacterial agent also had a significant effect on the SPAD of rice in all periods, and the application of a straw-degrading agent slowed down the senescence process of rice leaves and prolonged the photosynthesis functioning period of rice leaves, which laid the material foundation for the high yield of rice. The results suggested that the application of ZJW-6 was beneficial to root growth and development. This study also showed that 1 year of straw return reduces the number of grains per spike in rice, which results in lower yields. However, after the addition of ZJW-6, the yield of rice of S2 had significantly increased (Appendix A). Furthermore, the growth of rice was also greatly improved with S2 when compared with S0 and S1. The application of ZJW-6 was highly effective at increasing straw decomposition and crop growth (Appendix A, Figure 1). Therefore, it could be a desirable agronomic practice in rice fields when considering the dual goal of accelerating straw decomposition and offsetting the inhibition in growth induced by straw decomposition.

During the process of straw decomposition, microbial activity will compete with crop growth for soil N, thus resulting in a deficiency in soil N and the inhibition of crop growth, particularly in the case of accelerated straw decomposition [33,34]. However, in this study, the S2 treatments not only accelerated the rate of straw decomposition, promoted the release of nutrients from straw, and improved the soil fertility but also improved the tiller number, aboveground biomass, leaf area, and root growth compared with S0 (Figure 1). These results indicated that the straw-degrading bacterial agents could alleviate the inhibition of growth induced by straw decomposition. In this study, straw decomposition and crop growth performed best with S2, which indicated that the combined application of ZJW-6 substantially improved the balance between the nutrient supply in the soil and the combined nutrient demand of straw decomposition and crop growth, which favored high crop productivity.

Returning straw to the field may be an effective means to promote sustainable soil productivity, and the amount of soil enzymatic activity is widely used as a biological indicator of soil function and represents the driver of soil nutrient cycling [35,36]. The activities of soil enzymes, such as URE, ACP, and SUC, are sensitive to soil physicochemical changes because they provide information on the ability of soils to perform biogeochemical reactions [37]. URE promotes the mineralization of soil organic N to ammonium N, which can be adsorbed by soil particles and thus become easily accessible to plants [38]. ACP hydrolyzes organic compounds that contain P into inorganic P, which is required by plants [39]. SUC can enable the hydrolysis of sucrose to glucose and fructose as available energy matter for plants and soil microorganisms [40]. Not surprisingly, the activities of ACP, SUC, and URE in the soil were considerably higher in the S2 treatments than in the S1 treatments (Figure 6). This indicated that the soil enzymatic activity was heavily induced by straw returning, and the application of straw-degrading bacterial agents could definitely accelerate the rate of decomposition and, in turn, increase enzymatic activity, which could be due to the elevated metabolism of soil microorganisms and the stimulation of microbial activity by the addition of straw-degrading bacteria, as well as changes in the composition of microbial community. Therefore, straw returning with straw-degrading bacterial addition may increase the content of soil C and nutrients and create energy and a suitable environment for the activities of soil enzymes.

The composition of the microbial community controls the decomposition of organic matter, nutrient cycling, and rapid responses to changes in the soil environment [37,41]. Maintaining high levels of microbial diversity in the soil is vitally important for sustainable agriculture, which is becoming a key issue in developing sustainable productivity systems [42,43]. In addition, the loss of microbial diversity is the general consequence of long-term chemical fertilization, while returning crop residues could mitigate the negative effects from chemical fertilization on bacterial diversity [44,45]. Straw returning changed the abundance and diversity of soil bacteria, particularly with the addition of straw-degrading bacteria in this study. However, straw returning and the addition of degradative bacteria significantly reduced the abundance and diversity of soil fungi, respectively. We hypothesize that the addition of straw increased the burden of microbial decomposition, thus reducing the fungal diversity. In this study, we found that straw returning changed the structural composition of bacterial and fungal communities in soil, such as the relative abundances of Bacteroidota, Acidobacteriota, and Chlorofexi, which all changed significantly. These phenomena indicated that straw-degrading bacteria had an effect on microbial composition. The different changes of bacterial and fungal fractions indicated that microbes played important roles in straw decomposition and enzyme activities.

## 4. Materials and Methods

### 4.1. Sample Source

This experiment was conducted in 2021 and 2022 at an experimental site located in the National Crop Variety Approval and Characterization Station at Changchun County, Jilin Province, China (125°24′ E, 43°48′ N). This site is located in a temperate continental monsoon climate. The cumulative temperature and rainfall for the whole reproductive period were 2991 °C and 860.3 mm in 2021, respectively, and 2800 °C and 600 mm in 2022, respectively. Prior to the trial, the 0–20 cm soil of this region had an organic matter content of 16.55 g·kg^−1^, alkaline-dissolved N of 33.89 mg·kg^−1^, available P (AP) of 29.42 mg·kg^−1^, available K (AK) of 137.09 mg·kg^−1^ and pH of 6.70. The variety used was the laboratory conserved rice variety Jinongda 667 (Validation No.: JI Audited Rice 20190008).

Straw samples were taken from the experimental base of Jilin Agricultural University. After the crop had been mechanically harvested, straw pieces that were approximately 5–7 cm long were screened and sampled, subsequently air-dried, and stored for backup.

The straw-degrading bacterial agent was Iranian *Cellulomonas*, which was screened from the soil by our laboratory and patented (ZL 2020 1 1521445.6).

Three treatments were established in this study, including no straw return to the field (S0), straw returned to the field (S1), and straw returned + ZJW-6 (S2).

### 4.2. Experimental Design and Management

For the 2021 potting experiment, seedling pots (32 cm in diameter and 24 cm high) were each filled with 10 kg of soil. A total of 45 g of corn straw was placed at a depth of 10 cm in each pot. The straw was packed in gauze bags with 1 mm diameter holes for easier collection, and 15 g of straw was placed in each gauze bag. The bags were sealed with a sealer, and three packets per pot were horizontally buried in the soil. We sought to reflect field conditions as closely as possible, and the treatments purposefully did not represent unrealistic “extreme” conditions. The straw-degrading bacterial agent (ZJW-6) was applied at 60 g·667 m^−2^, and the fungicide and straw were mixed with the soil and packed into seedling pots.

In the 2022 field trial, each treatment was 30 m^2^, and 4.50 t·hm^−2^ of straw was returned to the field. Before the rice was returned to the field, the rice harvested in the previous season was threshed and crushed with a shredder to be 5–7 cm long. The crushed straw was spread evenly with the fungicide and then turned into the soil layer of 0-20 cm with a rotary tiller.

Each treatment was fertilized with 175 kg·hm^−2^ of pure N, which was applied as basal fertilizer/tiller fertilizer/spike fertilizer (6:3:1, *v*/*v*/*v*). There was a single application of 75 kg·hm^−2^ each of phosphate and potash fertilizer as a one-time basal fertilizer. The N fertilizer was urea (46% N); the P fertilizer was 12% P_2_O_5_; and the potash fertilizer was 60% K_2_O.

### 4.3. Measurement of the Parameters That Rice Growth under Field Conditions

The promotion of the growth of rice after straw incorporation with exogenous degrading bacteria was assessed. The experiment was conducted in a completely randomized design with three replicates. The agronomical parameters (root length, root number, root volume, photosynthetic characteristics and leaf area) were recorded.

Root system index: The root samples were collected, and the root volume was measured following the drainage method [46]. The length of the longest root and the number of roots per hole were measured. Three repetitions per treatment were used for the measurements, and the average value was calculated.

Leaf photosynthetic parameter: An Li-6400XT instrument (LI-6400, LI-COR, Inc., Lincoln, NE, USA) was used to measure the photosynthetic parameters with a built-in fixed light source and a light quantum density setting of 1200 µmol·m^−2^·s^−1^. Five representative plants per treatment were selected for the analysis. The photosynthetic rate was measured between 9:00 and 11:30 in the morning on a clear and windless day, with three replicates per treatment, and the mean values were calculated [47].

Measurement of the leaf area: Leaves from the representative plants were randomly collected in each plot at the tillering, jointing, full heading, and filling stages. The leaf area was determined using ImageJ 1.51j8 (Wayne Rasband, NIH, Bethesda, MD, USA). 

### 4.4. Measurement of the Tiller Number and Aboveground Biomass

Tiller number: After the rice seedlings had been transplanted, 10 representative plants (the plants and their surrounding plants grown normally) were randomly chosen in each plot to count the variation of tiller number every 7 days. 

Aboveground biomass: At the tillering, jointing, full heading, filling, and mature stages, the aboveground biomass was randomly sampled from the representative plants in each plot and weighed after oven-drying at 80 °C until a constant weight was achieved.

Grain yield measurement: At maturity, the number of effective spikes per hill in a representative 1 m^2^ plot was counted, and plants from five hills with the same average number of panicles were dried and tested to determine the number of effective panicles, spikelets per panicle, 1000-grain weight, and filled-grain rate. Each plot was harvested singly, and the quality and moisture content of the rice were measured after drying. The grain yield was obtained by converting the obtained measurements with a standard moisture content of 13.5%.

Measurement of the yield and yield components: Plants from the central 1 m^2^ in each plot were collected to determine the number of panicles, the number of spikelets per panicle, filled-grain percentage, 1000-grain weight, and grain yield at the time of harvest each year.

### 4.5. Determination of the Rate of Degradation 

The contents of cellulose, lignin, and hemicellulose in the straw were determined using the VanSoest washing method. Three previously buried nylon mesh bags that contained straw were removed from each treatment at the tillering, full heading and maturity stages, respectively, and the weight of the remaining straw was determined after washing and drying. Each treated straw sample was placed into a sieve (60 mesh) and rinsed with running water to ensure that the straw was not lost, before being dried to determine the rate of degradation, as follows [48]: Rate of straw degradation (%) = (W0 − W1)/W0 × 100%
where W0 is the mass of rice straw before degradation (g), and W1 is the mass of residual straw after degradation (g). All the experiments were repeated three times.

### 4.6. Electron Microscopy Observations

The straw samples with different treatments were taken, cleaned, placed in 2.5% glutaraldehyde, and fixed at 4 °C for more than 8 h. The samples were then dried in a desiccator. After cleaning by PBS and dehydration by an ethanol gradient, the samples were placed in a desiccator for drying and then sequentially placed on the scanning electron microscope (SEM) carrier stage with conductive adhesive tape, sprayed with gold, and placed under a field emission SEM (Hitachi SU 8010, Hitachi, Tokyo, Japan) at a voltage of 15 kV for observation and photography.

### 4.7. Measurements of Soil Enzymes Activities 

In each plot, topsoil (0–20 cm) samples were collected in September 2021 that were composed of five subsamples. The fresh soil was gently crushed and passed through a 2 mm sieve. The samples for DNA extraction were stored at −80 °C. The samples that were used for the analyses of soil enzyme activity were stored at 4 °C. The samples used for the analyses of soil major nutrients in the soil were air-dried and stored at room temperature. Samples of the soil were ground to <0.1 mm to enable the measurements of organic C using the potassium dichromate volumetric method in conjunction with the external heating method [49]. The soil total nitrogen (TN) was determined by the Kjeldahl method. The content of soil AP was measured by treatment with 0.5 mol L^−1^ sodium bicarbonate (NaHCO_3_) followed by molybdenum blue colorimetry [50]. The soil AK was extracted with 1 mol L^−1^ ammonium acetate (NH_4_OAc) (soil/solution, 1:10) and determined using a flame photometer [49]. The activities of the soil enzymes were assayed with kits as described by Bao (2000) [49]. The activity of soil urease (URE) was assayed using sodium phenol-sodium hypochlorite colorimetry; catalase (CAT) was assayed using potassium permanganate volumetric; soil acid phosphatase (ACP) was assayed using sodium phenyl phosphate colorimetry [49]; and soil sucrase (SUC) was assayed using 3, 5-dinitrosalicylic acid. 

### 4.8. Analysis of the Microbial Community Structure

The genomic DNA of the microbial communities was extracted from the soil samples using a Magnetic Soil DNA Kit (RT405-02; TIANGEN, Beijing, China). The DNA extract was checked using 1% agarose gel electrophoresis, and the concentration and purity of the DNA were determined using a NanoDrop 2000 (Thermo Fisher Scientific, Waltham, MA, USA). The hypervariable region V4 of the bacterial 16S rRNA gene was amplified with the primer pair 515F (5′-ACTCCTACGGGAGGCAGCAG-3′) and 806R (5′-GGACTACHVGGGTWTCTAAT-3′). The hypervariable region ITS1 of the fungi ITS rRNA gene was amplified with the primers ITS1 (5′-CTTGGTCATTTAGAGGAAGTAA-3′) and ITS2R (5′-GCTGCGTTCTTCATCGATGC-3′) using 10 ng of DNA template, 0.2 µM of forward and reverse primers, and 15 µL of Phusion^®^ High-Fidelity PCR Master Mix (New England Biolabs, Ipswich, MA, USA). Thermal cycling consisted of initial denaturation at 98 °C for 1 min, followed by 30 cycles of denaturation at 98 °C for 10 s, annealing at 50 °C for 30 s, and elongation at 72 °C for 30 s and 72 °C for 5 min. Paired-end sequencing was performed on a NovaSeq 6000 platform (Novogene Technology Co., Ltd., Tianjin, China) following the manufacturer’s instructions and clustered into operational taxonomic units (OTUs) at the 97% threshold. 

### 4.9. Bioinformatics Analysis and Statistics of the High-Throughput Sequencing Data

First, --FLASH (v. 1.2.11, http://ccb.jhu.edu/software/FLASH/ accessed on 19 June 2024) [51] was used to merge paired-end sequences of the sequencing data and rename them. FASTP v. 0.23.1 was used to obtain high-quality Raw Tags. Clean Tag was used to remove the double-ended primers and bar codes and to perform quality control to ensure that the error rate was <1% [52]. The --derep_fulllength command was used to reduce sequence redundancy. Denoising was performed with DADA2 or deblur module in QIIME2 (v. QIIME2-202202) to obtain the initial Amplicon Sequence Variants (ASVs). The species were annotated using QIIME2. To analyze the diversity, richness, and uniformity of the communities in the sample, the alpha-diversity was calculated from seven indices in QIIME2, including Observed_otus, Chao1, Shannon, Simpson, Dominance, Good’s coverage, and Pielou_e. The complexity of the community composition was evaluated, and the differences between samples (groups) was compared using beta-diversity and calculated based on weighted and unweighted UniFrac distances in QIIME2.

All the sequence data were deposited in the NCBI Sequence Read Archive database. The accession number is PRJNA1100920 for the bacterial data and PRJNA1100941 for the fungal data.

### 4.10. Statistical Analysis

All the experiments were conducted in triplicate (n = 3). A one-way analysis of variance (ANOVA) was performed to determine statistically significant differences between the control and other treatments using SPSS 20.0 (IBM, Inc., Armonk, NY, USA). The differences were considered statistically significant at *p* < 0.05. The data are presented as the mean ± SD of three replicates. Partial least-squares path modeling (PLS-PM) was conducted with Smart PSL v. 4.0 (Smart PLS GmbH, Böenningstedt, Germany) to further identify the possible pathways of exogenous degrading bacteria, straw degradation capacity, soil physical and chemical properties, and soil microbial community. All the figures were plotted using OriginPro 2017 (OriginLab, Northampton, MA, USA).

## 5. Conclusions

This study demonstrated that the rate of straw decomposition, crop growth (e.g., tiller number, aboveground biomass, leaf area, and root growth), and grain yield all improved with ZJW-6. It also enhanced the rate of rice straw degradation by inducing positive interactions between the bacterial and fungal genera and enriching the microbial taxa related to straw degradation. By contrast, the straw decomposes slowly during the early stage of returning to the field without ZJW-6, most of the nitrogen in the soil is scrambled by microorganisms, and the crop grows slowly due to the lack of nitrogen. This study promotes our understanding of the straw degradation mechanism at the molecular level, provides a scientific basis to explain the variation tendency of soil microorganisms after straw returning to the field, and offers promising insights into managing rice straw resources.

## Figures and Tables

**Figure 1 ijms-25-07835-f001:**
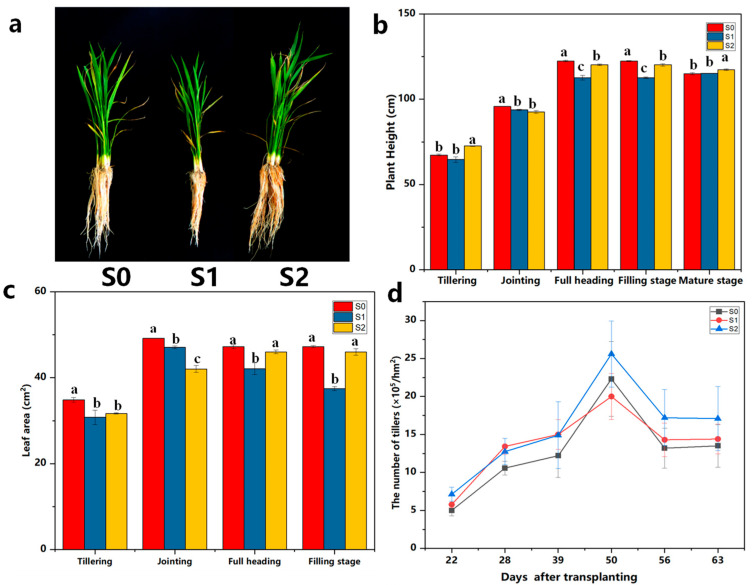
Effects of the addition of straw-degrading bacterial agents on the growth of rice. Phenotypes of rice (**a**); Plant height (**b**); Leaf area (**c**); Number of tillers (**d**); Dry matter accumulation (**e**); Root activity (**f**); Number of roots (**g**); Root length (**h**). Different lowercase letters indicate unified indicators that demonstrate significant differences between the S0, S1 and S2 treatments (*p* < 0.05), no straw return to the field (S0), straw returned to the field (S1), and straw returned + ZJW-6 (S2). Each experiment was repeated three times. Data are the mean ± SD of three biological replicates.

**Figure 2 ijms-25-07835-f002:**
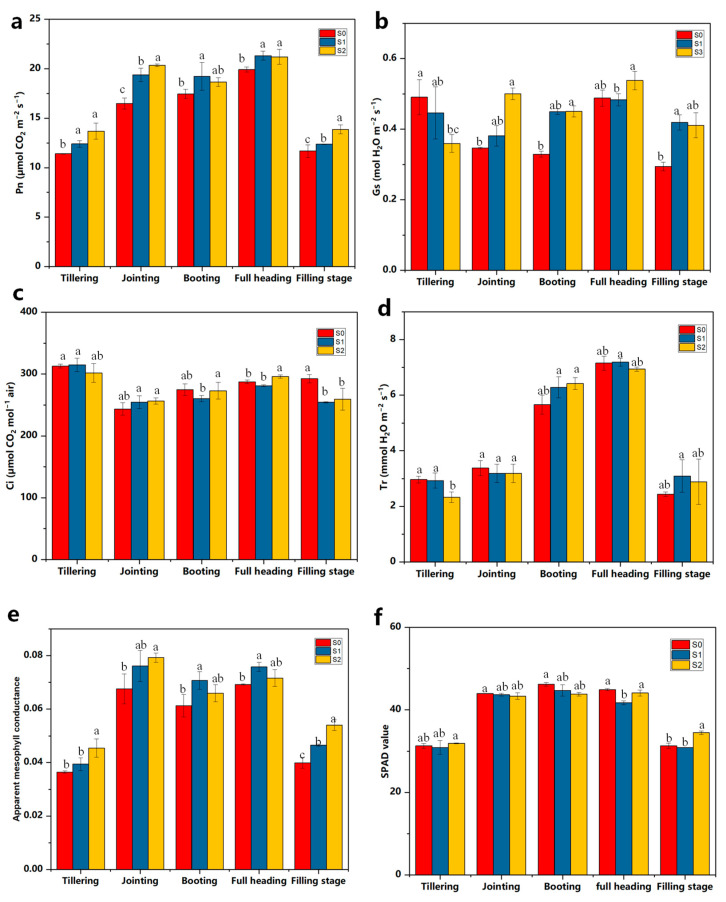
Effects of straw-degrading bacteria on photosynthetic characteristics of rice. Pn, net photosynthesis rate (**a**); Gs, stomatal conductivity (**b**); Ci, intercellular carbon dioxide concentration (**c**); Tr, transpiration rate (**d**); Apparent mesophyll conductance (**e**). SPAD value (**f**); Different lowercase letters indicate significant differences between S0, S1 and S2 at *p* < 0.05, no straw return to the field (S0), straw returned to the field (S1), and straw returned + ZJW-6 (S2).

**Figure 3 ijms-25-07835-f003:**
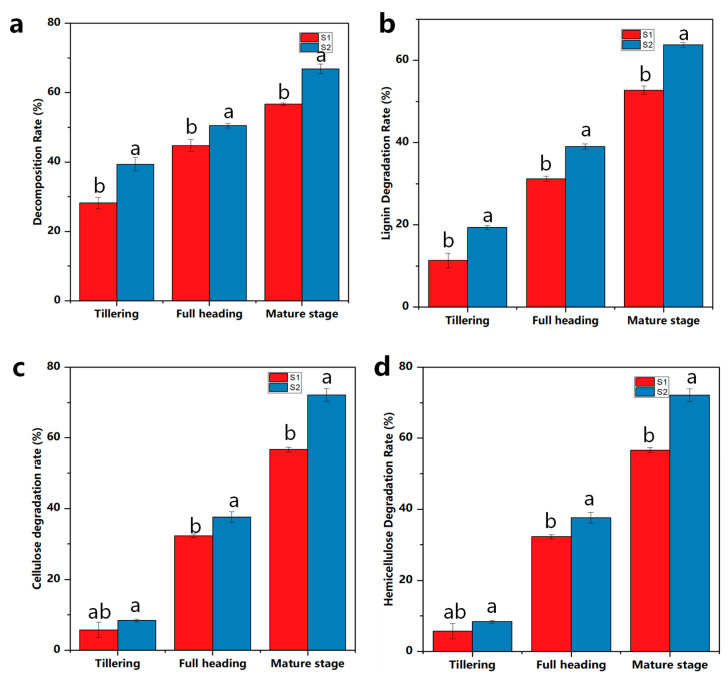
Effect of straw-degrading bacteria on straw degradation. Decomposition rate (**a**) and the lignin degradation rate (**b**), cellulose degradation rate (**c**), and hemicellulose degradation rate (**d**). Different lowercase letters indicate a significant difference between the S1 and S2 treatments at *p* < 0.05, straw returned to the field (S1), and straw returned + ZJW-6 (S2).

**Figure 4 ijms-25-07835-f004:**
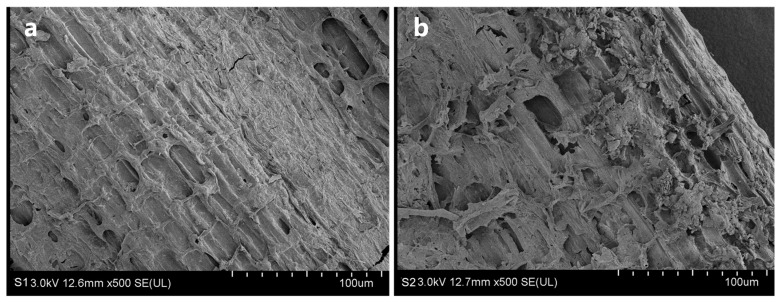
Scanning electron microscopic observation of the straw after degradation. Non-degrading bacteria treatment (**a**) and degrading bacteria treatment (**b**).

**Figure 5 ijms-25-07835-f005:**
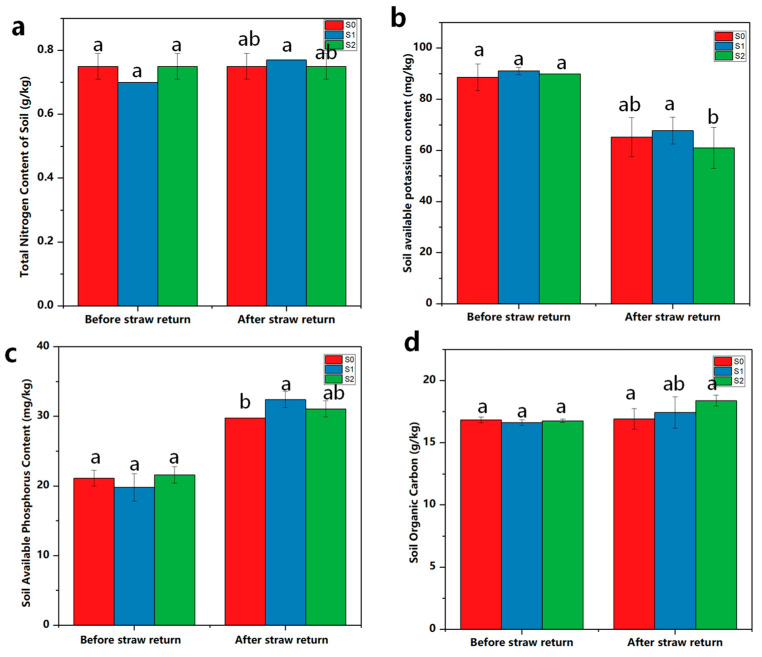
Effects of different treatments on the total nitrogen (**a**), soil available potassium content (**b**), soil available phosphorus content (**c**), soil organic carbon (**d**). SOC, soil organic carbon. Different lowercase letters indicate significant differences between S0, S1 and S2 at *p* < 0.05, no straw return to the field (S0), straw returned to the field (S1), and straw returned + ZJW-6 (S2).

**Figure 6 ijms-25-07835-f006:**
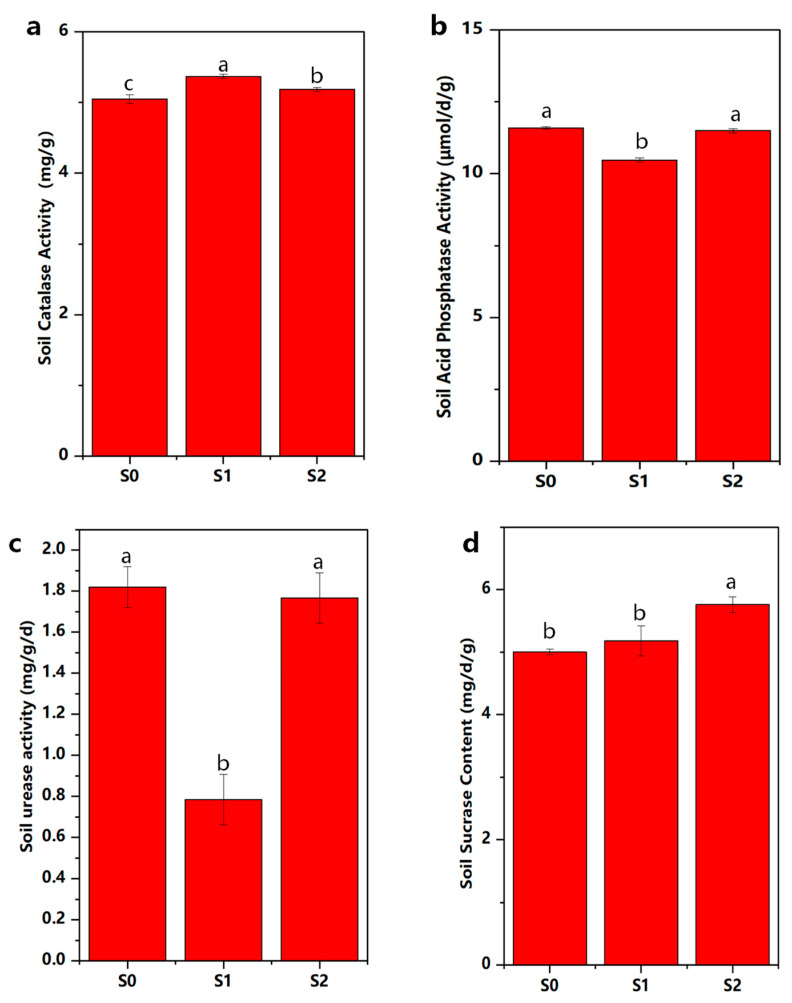
Enzyme activity of soil under different treatments. Acid phosphatase (**a**), Catalase (**b**), Urease (**c**), Sucrase (**d**). The error bars represent the standard errors. Different lowercase letters indicate significant differences between S0, S1 and S2 at *p* < 0.05, no straw return to the field (S0), straw returned to the field (S1), and straw returned + ZJW-6 (S2).

**Figure 7 ijms-25-07835-f007:**
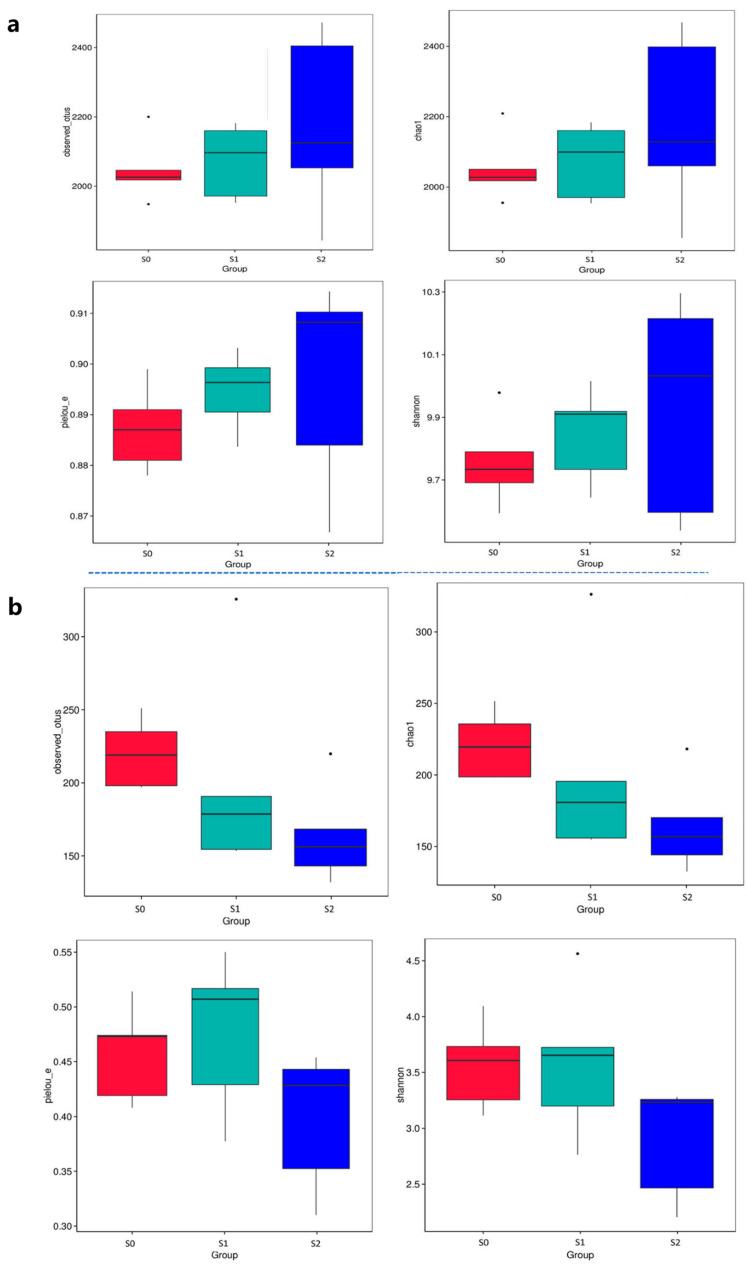
α-diversity of bacteria and fungi among the three treatments. α-diversity of the bacteria in the soil (**a**). α-diversity of the fungi in the soil (**b**).

**Figure 8 ijms-25-07835-f008:**
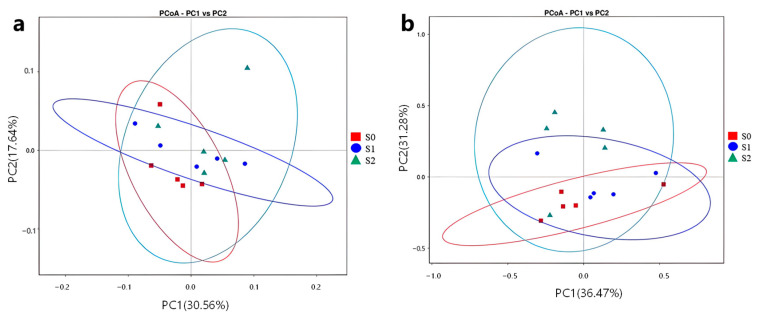
Unconstrained principal coordinate analysis (PCoA) with a weighted UniFrac distance on the beta-diversity of the soil bacterial (**a**) and (**b**) fungal communities.

**Figure 9 ijms-25-07835-f009:**
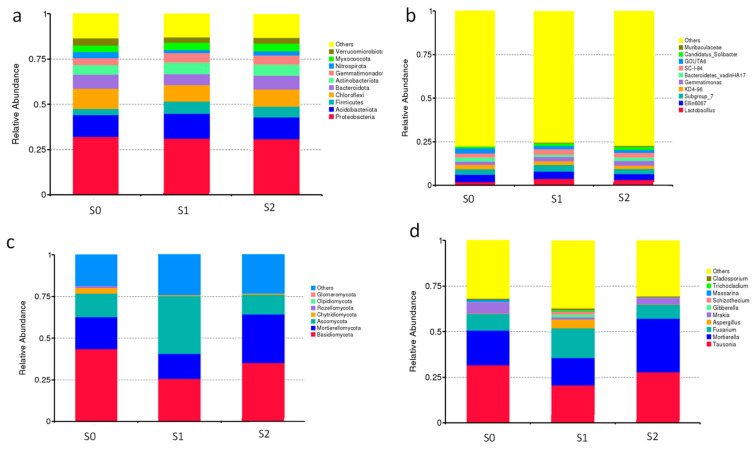
Relative abundances of the bacteria and fungi at the phylum level (**a**,**c**) and genus level (**b**,**d**) with statistical differences in each treatment.

**Figure 10 ijms-25-07835-f010:**
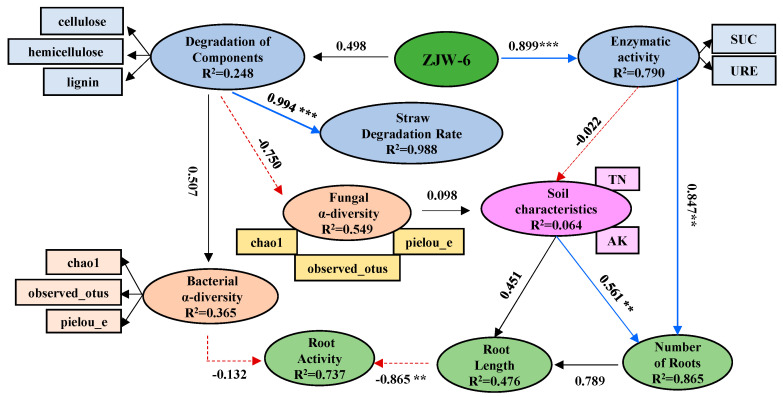
Partial least-squares path model, which indicates the effects of exogenous degradative bacteria and the growth of roots on the soil microbial properties via soil physicochemical properties and vegetative parameters. Blue arrows, significant (*p* < 0.01) positive paths; Red lines, non-significant (*p* > 0.05) pathways. Numbers within the circles, R2 of the corresponding parameters; numbers near the lines, the path coefficients; AK, available potassium; TN, total nitrogen; SUC, sucrase; URE, urease. (** *p* < 0.01. *** *p* < 0.001).

## Data Availability

The study’s original contributions are included in the article/Appendix A; additional inquiries should be addressed to the corresponding authors.

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
