# Peer review of "Straw Incorporation with Exogenous Degrading Bacteria (ZJW-6): An Integrated Greener Approach to Enhance Straw Degradation and Improve Rice Growth"

_ijms, 2024, doi:10.3390/ijms25147835_

Round 1

Reviewer 1 Report

Comments and Suggestions for Authors

Dear Authors, Editors,

Thank you for the opportunity to review this paper. I find it interesting and important for the scope of this journal. However, I have few comments, which I believe may improve the quality of this work.

1. I personally find that the material and methods section appearing before results and discussion is a clearer way of presentation, and it could increase understanding of the experiment

2. I note the need for greater clarification of the mechanisms of action with exogenous degrading bacteria behind the outcomes presented in the results section.

3. Result section is very comprehensive with comparison to the discussion section. It might make reading the article difficult. Try to transfer some data to the supplementary materials, and enhance discussion section. 

4. It would be beneficial to describe possible future research regarding exogenous degrading bacteria, and present advantages and drawbacks of large-scale applications of them. What would the process of implementing such a solution look like?

5. Please re-read the article to find some typos and improve editing. Examples: line 58- unnecessary space after reference; line 61- weird coma between references; Figure 1g,h visibility could be improved; lines 252-255- please add descriptions when abbreviations appare for the first time; line 463 (material and methods)-  standardize the notation of units; line 555- recheck the chapter title

Author Response

Point-to-point response to reviewers

Manuscript Title: Straw incorporation with exogenous degrading bacteria (ZJW-6): An integrated greener approach to Enhance Straw Degradation and improve rice growth

Dear Editor:

First of all, we deeply appreciate the editor for your time and efforts. Based on your comments, we have uploaded the data in the manuscript and corrected the relevant experimental details. The revised parts are marked in different colors in the manuscript. Our specific response is as follows.

Reviewer 1

Thank you for the opportunity to review this paper. I find it interesting and important for the scope of this journal. However, I have few comments, which I believe may improve the quality of this work.

  1. I personally find that the material and methods section appearing before results and discussion is a clearer way of presentation, and it could increase understanding of the experiment. 

Response: We appreciate the reviewer’s kind suggestions, we have reviewed many articles in IJMS journals and the materials and methods are ranked after the results, so this chapter is also laid out according to the IJMS journals.

  1. I note the need for greater clarification of the mechanisms of action with exogenous degrading bacteria behind the outcomes presented in the results section.

Response: Thank you for your kindly suggestion. In the article, the effects of the addition of straw-degrading bacteria on straw degradation, soil nutrients and microbiological changes were investigated. the addition of ZJW-6 affected the growth by indirectly regulating the soil microbial characteristics through their effects on straw decomposition and the soil physical and chemical properties. The results showed that ZJW-6 not only directly participated in straw degradation, but also stimulated the competitions of bacterial communities, which indirectly promoted straw degradation. The ZJW-6 also increased the root biomass by in-creasing the rate of degradation and activity of the soil enzymes, which alleviated negatively affected the effect of straw returning on plant growth, providing favorable conditions for rice growth, promoting its growth and development. A discussion of the mechanism of action of exogenous degrading bacteria has been discussed in Result 2.10.

  1. Result section is very comprehensive with comparison to the discussion section. It might make reading the article difficult. Try to transfer some data to the supplementary materials, and enhance discussion section. 

Response: Thank you very much for your useful comments. We transferred the yield data from the results to the supplementary material and enhanced the discussion section.

  1. It would be beneficial to describe possible future research regarding exogenous degrading bacteria, and present advantages and drawbacks of large-scale applications of them. What would the process of implementing such a solution look like?

Response: Thank you very much for your useful comments. More in-depth studies on the degradation mechanisms of exogenous degrading bacteria will be carried out in the future and these bacteria will be applied on a large scale. This is because the use of straw return is not only an effective management practice to improve soil fertility in agro-ecosystems, but also nurtures environmental stewardship by reducing waste and carbon footprint. Screening microorganisms with the ability to degrade straw is an effective way to solve the problem of fresh straw not being able to decompose quickly after being returned to the field, and it is of great significance for farmers to accelerate the promotion of crop straw return to the field and realise the sustainable use of agricultural resources.

  1. Please re-read the article to find some typos and improve editing. Examples: line 58- unnecessary space after reference; line 61- weird coma between references; Figure 1g, h visibility could be improved; lines 252-255- please add descriptions when abbreviations appare for the first time; line 463 (material and methods)- standardize the notation of units; line 555- recheck the chapter title

Response: Thank you very much for your useful comments. We have removed unnecessary spaces in line 58, please see L59; modified the symbol between the two documents in line 61, please see L62; modified Fig. 1g, H, please see Fig. 1g, H; and added descriptions to the abbreviations in lines 252-255, please see L240-242; The unit symbol in line 463(materials and methods) is modified, please see materials and methods; the section heading in line 555 is changed, please see L554.

Reviewer 2 Report

Comments and Suggestions for Authors

The work titled: "Straw incorporation with exogenous degrading bacteria (ZJW-6): An integrated greener approach to Enhance Straw Degradation and improve rice growth" addresses a very interesting topic. This topic concerns the rational management of rice straw, which is consistent with the assumptions of the currently widely promoted circular economy.

The summary provides a good introduction to the topic and contains the most important information regarding the assumptions and results of the research performed.

The sections in the manuscript are appropriately arranged, which is in line with the publisher's requirements. The authors took care to ensure the readability of the figures and tables.

A very beneficial and enriching article is Figure 4. Scanning electron microscopic observation of the straw after degradation, which very well illustrates the degree of degradation of rice straw.

Necessary explanations are provided under the tables and figures.

The results are quite detailed but clearly described. The discussion of the obtained results explains the obtained results and the presented relationships quite well.

The conclusions well reflect the entire work and the results achieved.

However, after carefully reading the entire work, a few suggestions come to my mind that I propose to consider:

1. Introduction. Lines 87 – 97 require correction. I propose to move some of this information, regarding information about the treatments performed and their markings, to section 5.2. Experimental design and management. In this paragraph (lines: 87-97), the research hypothesis and the purpose of the research should be more emphasized. However, I propose to include the results and information about the usefulness of the results (lines: 94-97) in section 4. Conclusions.

2. RESULTS

- Figure 2d and 2F, letters are missing above some bars,

- Figure 3 – lines 214-215 – description does not fully correspond to the drawing, please correct the caption or data in the drawing.

- Figure 8 – a slightly better quality of the drawing would be useful.

3. Discussion - here it would be good to additionally refer to the latest research results (2020-2024), because the literature used in the discussion concerns the period 2000-2019.

4. Methodology.

- section 5.1. – it would be better to provide the temperature value as the average temperature per day, possibly with minimum and maximum values ​​throughout the entire period.

 - Section 5.2., line 496, please provide chemical formulas of fertilizers or their names in brackets. The current record is misleading.

5. References – line 754 – please correct the entry in accordance with editorial requirements.

In further research, in my opinion, a more modern method of determining carbon content should be considered. Because the method described in the methodology uses a carcinogenic reagent, its use should be very limited.

After making the necessary corrections, in my opinion the work can be accepted for publication in the International Journal of Molecular Sciences.

Author Response

Comments and Suggestions for Authors

The work titled: "Straw incorporation with exogenous degrading bacteria (ZJW-6): An integrated greener approach to Enhance Straw Degradation and improve rice growth" addresses a very interesting topic. This topic concerns the rational management of rice straw, which is consistent with the assumptions of the currently widely promoted circular economy.

The summary provides a good introduction to the topic and contains the most important information regarding the assumptions and results of the research performed.

The sections in the manuscript are appropriately arranged, which is in line with the publisher's requirements. The authors took care to ensure the readability of the figures and tables.

A very beneficial and enriching article is Figure 4. Scanning electron microscopic observation of the straw after degradation, which very well illustrates the degree of degradation of rice straw.

Necessary explanations are provided under the tables and figures.

The results are quite detailed but clearly described. The discussion of the obtained results explains the obtained results and the presented relationships quite well.

The conclusions well reflect the entire work and the results achieved.

However, after carefully reading the entire work, a few suggestions come to my mind that I propose to consider:

  1. Introduction. Lines 87 – 97 require correction. I propose to move some of this information, regarding information about the treatments performed and their markings, to section 5.2. Experimental design and management. In this paragraph (lines: 87-97), the research hypothesis and the purpose of the research should be more emphasized. However, I propose to include the results and information about the usefulness of the results (lines: 94-97) in section 4. Conclusions.

Response: We appreciate the reviewer’s kind suggestions, we have moved the information on the treatments performed and their labelling to Materials and Methods, please see L476-477. Also include information related to the results in Conclusions.

  1. RESULTS

- Figure 2d and 2F, letters are missing above some bars,

Response: We appreciate the reviewer’s kind suggestions, we have added the letters above the bars in Figures 2d and 2F.

- Figure 3 – lines 214-215 – description does not fully correspond to the drawing, please correct the caption or data in the drawing.

Response: We appreciate the reviewer’s kind suggestions, we have re-described Figure 3, In order to make the description exactly the same as the drawing.

- Figure 8 – a slightly better quality of the drawing would be useful.

 Response: We appreciate the reviewer’s kind suggestions, we processed Figure 8 to make it look a little better.

  1. Discussion - here it would be good to additionally refer to the latest research results (2020-2024), because the literature used in the discussion concerns the period 2000-2019.

 Response: We appreciate the reviewer’s kind suggestions, We have added new research results (2020-2024) to the discussion.

  1. Methodology.

- section 5.1. – it would be better to provide the temperature value as the average temperature per day, possibly with minimum and maximum values ​​throughout the entire period.

Response: We appreciate the reviewer’s kind suggestions, We only focused on the cumulative temperature throughout the remaining period of the study because all treatments were performed under the same conditions, so we did not record the average temperature for each day, and we will pay more attention to it in the subsequent experiments.

 - Section 5.2., line 496, please provide chemical formulas of fertilizers or their names in brackets. The current record is misleading.

Response: We appreciate the reviewer’s kind suggestions, we have re-described it in the article, please see L495-495.

  1. References – line 754 – please correct the entry in accordance with editorial requirements.

 Response: We appreciate the reviewer’s kind suggestions, we have corrected the entry in accordance with editorial requirements.

In further research, in my opinion, a more modern method of determining carbon content should be considered. Because the method described in the methodology uses a carcinogenic reagent, its use should be very limited.

 Response: We appreciate the reviewer’s kind suggestions, In further studies, we will consider more modern methods to determine the carbon content, thank you again.

After making the necessary corrections, in my opinion the work can be accepted for publication in the International Journal of Molecular Sciences.

Reviewer 3 Report

Comments and Suggestions for Authors

The manuscript titled "Straw incorporation with exogenous degrading bacteria (ZJW-6): An integrated greener approach to Enhance Straw Degradation and improve rice growth" contains interesting research results for science and agricultural practice.

I read the manuscript with interest. I appreciate that this is a field experiment, but it is better to do two or three year field experiments. I included my comments in the original PDF text.

I hope that my comments will help the authors improve the text. After making corrections, I recommend publishing the manuscript in the IJMS journal.

General notes:

correct author affiliations

line 29. write Latin names in italics

add to keywords: "physiological measurements"

line 63. correct citations

In the Materials and Methods, describe in detail the studied factors S0 S1 and S2

Correct the description of figure 1

Improve figure stats 1h

line 165 explain the abbreviation SPAD

correct the description of figure 2

check the units according to the journal requirements, e.g. table 1

correct the description of figure 3

correct the statistics in Figure 5a

correct the description of figures 6, 7 and 8

line 311. correct it and complete it in the literature list

Describe this subsection in the Discussion: 2.2. Effects of the addition of straw-degrading bacteria on the photosynthetic capacity of rice

Place the Conclusion chapter at the end of the manuscript

If you have weather conditions in the year of research, please provide them

In which laboratory was the soil analysis performed (accredited)

Describe what variety of rice you tested

What was the forecrop in your experiment

line 594 correct it

Best regards

Author Response

Comments and Suggestions for Authors

The manuscript titled "Straw incorporation with exogenous degrading bacteria (ZJW-6): An integrated greener approach to Enhance Straw Degradation and improve rice growth" contains interesting research results for science and agricultural practice.

I read the manuscript with interest. I appreciate that this is a field experiment, but it is better to do two or three year field experiments. I included my comments in the original PDF text.

I hope that my comments will help the authors improve the text. After making corrections, I recommend publishing the manuscript in the IJMS journal.

General notes:

  1. correct author affiliations

Response: We appreciate the reviewer’s kind suggestions, we have correct author affiliations.

  1. line 29. write Latin names in italics

Response: We appreciate the reviewer’s kind suggestions, we have written Latin names in italics, please see L29.

  1. add to keywords: "physiological measurements"

Response: We appreciate the reviewer’s kind suggestions, we have added to keywords: "physiological measurements", please see L38.

  1. line 63. correct citations

Response: Thank you very much for your useful comments, we have corrected it. please see L66.

5.In the Materials and Methods, describe in detail the studied factors S0 S1 and S2

Response: Thank you very much for your useful comments, we have described in detail the studied factors S0 S1 and S2 in the Materials and Methods, please see L476-477.

  1. Correct the description of figure 1

Response: Thank you very much for your useful comments, we have corrected the description of figure 1, please see figure 1.

  1. Improve figure stats 1h

Response: Thank you very much for your useful comments, we have improved figure stats 1h, please see figure 1h.

  1. line 165 explain the abbreviation SPAD

Response: Thank you very much for your useful comments, we have explained the abbreviation SPAD, please see L163.

  1. correct the description of figure 2

Response: Thank you very much for your useful comments, we have corrected the description of figure 2.

  1. check the units according to the journal requirements, e.g. table 1

Response: Thank you very much for your useful comments, we have checked and corrected the units in Table 1 as required by the journal and moved Table 1 to the supplementary material

  1. correct the description of figure 3

Response: Thank you very much for your useful comments, we have corrected it.

  1. correct the statistics in Figure 5a

Response: Thank you very much for your useful comments, we have corrected it.

  1. correct the description of figures 6, 7 and 8

Response: Thank you very much for your useful comments, we have corrected the description of figures 6, 7 and 8.

  1. line 311. correct it and complete it in the literature list

Response: Thank you very much for your useful comments, we have added the reference it in the literature list.

  1. Describe this subsection in the Discussion: 2.2. Effects of the addition of straw-degrading bacteria on the photosynthetic capacity of rice

Response: Thank you very much for your useful comments, we have described 2.2 subsection in the discussion, please see L379-388.

  1. Place the Conclusion chapter at the end of the manuscript

Response: Thank you very much for your useful comments, according to the IJMS journal requirements, Materials and Methods is the final part of the article.

  1. If you have weather conditions in the year of research, please provide them

Response: Thank you very much for your useful comments, The article focuses on the effects of the addition of exogenous degrading bacteria on straw degradation and rice plant growth, and did not study the weather conditions of the year, but only recorded the cumulative temperature and rainfall of the year.

  1. In which laboratory was the soil analysis performed (accredited)

Response: We appreciate the reviewer’s kind suggestion, Soil analyses were carried out in the laboratory of the Crop Physiology and Cultivation Team of Jilin Agricultural University (JLAU).

  1. Describe what variety of rice you tested

Response: We appreciate the reviewer’s kind suggestions, Thank you very much for your useful comments. We used the rice variety Jinongda 667 (Validation No.: JI Audited Rice 20190008), which is the main cultivar in Jilin Province, we have added that description to the material approach of the article, please see L468-469.

  1. What was the forecrop in your experiment

Response: Thank you very much for your useful comments. Forecrop in my experiment is rice.

  1. line 594 correct it

Response: Thank you very much for your useful comments. We've corrected it, please see L593.